# Survival and lung function decline in patients with definite, probable and possible idiopathic pulmonary fibrosis treated with pirfenidone

**Ondřej Májek**[1], **Jakub Gregor**[1], **Nesrin Mogulkoć**[2], **Katarzyna Lewandowska**[3], **Martina Šterclová**[4], **Veronika Müller**[5], **Marta Hájková**[6], **Mordechai R. Kramer**[7], **Jasna Tekavec-Trkanjec**[8], **Dragana Jovanović**[9], **Michael Studnicka**[10], **Natalia Stoeva**[11], **Klaus-Uwe Kirchgässler**[12], **Simona Littnerová**[1], **Ladislav Dušek**[1], **Martina Koziar Vašáková**[4]*

**1** Institute of Biostatistics and Analyses, Faculty of Medicine, Masaryk University, Brno, Czech Republic, **2** Department of Pulmonary Medicine, Ege University Medical School, Izmir, Turkey, **3** 1st Department of Pulmonary Diseases, Institute of Tuberculosis and Lung Diseases, Warsaw, Poland, **4** Department of Respiratory Diseases of the First Faculty of Medicine Charles University, Thomayer University Hospital, Prague, Czech Republic, **5** Department of Pulmonology, Semmelweis University, Budapest, Hungary, **6** Clinic of Pneumology and Phthisiology, University Hospital Bratislava, Bratislava, Slovakia, **7** Institute of Pulmonary Medicine, Rabin Medical Center, Petah Tikva, Israel, **8** Pulmonary Department, University Hospital Dubrava, Zagreb, Croatia, **9** Internal Medicine Clinic Akta Medica, Belgrade, Serbia, **10** Paracelsus Medical University, SALK, Salzburg, Austria, **11** Pulmonary Department, Acibadem City Clinic Tokuda Hospital, Sofia, Bulgaria, **12** F. Hofmann-La Roche Ltd, Basel, Switzerland

* martina.koziarvasakova@ftn.cz

**Data Availability Statement:** All relevant data are within the paper and its Supporting Information files.

## Abstract

### Background

There is no clear evidence whether pirfenidone has a benefit in patients with probable or possible UIP, i.e. when idiopathic pulmonary fibrosis (IPF) is diagnosed with a lower degree of diagnostic certainty. We report on outcomes of treatment with pirfenidone in IPF patients diagnosed with various degrees of certainty.

### Methods and findings

We followed patients in the multi-national European MultiPartner IPF Registry (EMPIRE) first seen between 2015 and 2018. Patients were assessed with HRCT, histopathology and received a multi-disciplinary team (MDT) IPF diagnosis. Endpoints of interest were overall survival (OS), progression-free survival (PFS) and lung function decline.

### Results

A total of 1626 patients were analysed, treated with either pirfenidone (N = 808) or receiving no antifibrotic treatment (N = 818). When patients treated with pirfenidone were compared to patients not receiving antifibrotic treatment, OS (one-, two- and three-year probability of survival 0.871 vs 0.798; 0.728 vs 0.632; 0.579 vs 0.556, P = 0.002), and PFS (one-, two- and three-year probability of survival 0.597 vs 0.536; 0.309 vs 0.281; 0.158 vs 0.148, P =

**Funding:** This study was supported by F. Hofmann-La Roche. The EMPIRE registry is supported by Boehringer Ingelheim and F. Hofmann-La Roche. The funders had no role in study design, data collection and analysis, decision to publish, or preparation of the manuscript.

**Competing interests:** Dr. Kirchgässler is an employee of F. Hoffmann-La Roche Ltd. Dr. Lewandowska reports grants, personal fees and non-financial support from F. Hofmann-La Roche, grants, personal fees and non-financial support from Boehringer Ingelheim. Dr. Müller received grants and personal fees from Roche and Boehringer Ingelheim. Dr. Stoeva reports and received fees from Boehringer Ingelheim for participation in advisory boards and from Roche for presentations. Dr. Studnicka reports grants and personal fees from Boehringer-Ingelheim, grants and personal fees from Roche, during the conduct of the study. Dr. Tekavec-Trkanjec reports personal fees from Roche, personal fees and other from Boehringer Ingelheim. This does not alter our adherence to PLOS ONE policies on sharing data and materials.

0.043) was higher, and FVC decline smaller (-0.073 l/yr vs -0.169 l/yr, P = 0.017). The benefit of pirfenidone on OS and PFS was also seen in patients with probable or possible IPF.

## Conclusions

This EMPIRE analysis confirms the favourable outcomes observed for pirfenidone treatment in patients with definitive IPF and indicates benefits also for patients with probable or possible IPF.

## Introduction

Idiopathic pulmonary fibrosis (IPF) is characterized by progressive fibro-proliferative scarring of the lung parenchyma, originating from alveolar lesions. IPF is most frequently seen in predisposed middle-aged and older individuals, ultimately leading to respiratory failure and death.

Despite current treatment options with antifibrotics, the prognosis of IPF remains unfavourable. Two drugs, namely pirfenidone and nintedanib, have become available in the last decade, and have demonstrated to slow progression of IPF. However, antifibrotic treatment cannot reverse fibrotic lung remodelling.

Randomised clinical trials (RCT), followed by open extensions of these trials and post-hoc assessments, have documented treatment efficacy for these two drugs [1–8]. Patients participating in these RCTs were rather homogeneous with regard to age, comorbidity pattern and high-resolution computed tomography (HRCT) findings. As a consequence, real-world IPF patients very often do not match inclusion criteria of these RCTs [9]. One group of particular concern is IPF patients who do not present with definite usual interstitial pneumonia (UIP) on HRCT, but demonstrate the radiological and/or histopathological pattern of probable/ possible UIP.

The RCTs, which have led to the registration of pirfenidone, included either 1) patients with a definite UIP pattern on HRCT or 2) patients with only a probable/possible UIP on HRCT, but later confirmation of UIP by lung biopsy [10]. Despite the many studies, which have investigated pirfenidone, there is still lack of evidence as to whether pirfenidone can influence outcomes in patients with a probable/possible UIP pattern on HRCT, and no subsequent histopathological confirmation of UIP. In many elderly patients presenting with probable or possible UIP, the risk for diagnostic lung biopsy is often considered too high, thereby preventing possibly beneficial antifibrotic IPF treatment.

Real-world registries can be a valuable source of information to demonstrate the effectiveness of pirfenidone for these particular subgroups of patients [11, 12]. Furthermore, length of follow-up of RCTs is limited. When IPF registries run over several years, they can, in addition to RCTs, provide survival data for IPF subgroups for longer time periods.

Publications based on IPF real-world registries have recently described the epidemiology, the clinical management, and the treatment outcomes associated with IPF in various countries [13–18]. Although IPF registries usually follow a greater number of participants than RCTs, they can still be limited, because of small numbers of patients falling into specific sub-categories such as probable or possible UIP.

At the time of reporting, more than 3,500 IPF patients from 50 hospitals in eleven European and Middle Eastern countries have been enrolled into the EMPIRE registry [14, 19–21]. Based

on this large and multi-national IPF population, important new evidence for small subgroups of IPF patients can be generated.

In the present analysis, we report overall survival, progression-free survival and lung function decline for patients treated with pirfenidone according to subgroups of different diagnostic certainty as defined by HRCT pattern and histopathology.

## Methods

### Study population

This analysis includes IPF patients with their first visit to the EMPIRE registry taking place between 1 January 2015 and 31 December 2018 and followed through 29 October 2019.

IPF was diagnosed according to the 2011 ATS/ERS/JRS/ALAT criteria [22]. That is, all patients included in the EMPIRE registry were considered to have had HRCT, lung biopsy for histopathological assessment if indicated, and a MDT discussion for final IPF diagnosis. Final IPF diagnosis was always determined by a local multidisciplinary team (MDT); the EMPIRE data therefore also include patients with HRCT or histopathological findings not demonstrating UIP pattern.

The present data analysis included 1) patients treated with pirfenidone and 2) patients not treated with antifibrotics (herein referred as the no antifibrotic treatment group) and was done according to subgroups of diagnostic certainty (see below). Patients followed in EMPIRE and receiving nintedanib at any time were excluded from analysis (see S1 Fig).

For the purpose of longitudinal analysis, a participant's baseline visit (start of follow-up) was defined as follows: 1) patients treated with pirfenidone were analysed from the time onwards when treatment was started; 2) patients on no antifibrotic treatment were analysed from the time when the first visit to EMPIRE was recorded. End of follow-up was defined when death, lung transplantation or progression of disease (for PFS analysis) was observed, and was censored either at the date of the last visit to the registry, or the date of the last documented pirfenidone treatment (for the pirfenidone group) or the date when a patient was lost to follow-up. If a patient terminated the pirfenidone therapy, the subsequent treatment period (no antifibrotics) was not included in the analysis.

Study measurements (forced vital capacity–FVC, diffusing capacity for carbon monoxide–DLCO, and six-minute walk test– 6MWT) were collected at baseline and approximately every 6 months thereafter. However, the frequency of visits might fluctuate in accordance with local clinical practice or regulatory requirements for the monitoring of pirfenidone therapy.

All measurements were entered in a standardised way into a web-based data management tool with secure electronic data transfer. All data was cross-checked for plausibility (using in-built algorithms during data entry) and outlying observations. The study was approved by the Ethics Committee of Thomayer University Hospital and Institute for Clinical and Experimental Medicine, Prague, Czech Republic. The EMPIRE registry was approved by local ethics committees in individual countries and sites involved in the registry. All patients signed an informed consent prior to enrolment into the registry.

EMPIRE was set-up to reflect the real-world management of patients with IPF. IPF patients were enrolled by hospitals in European and Middle Eastern countries with a long-standing expertise in the management of interstitial lung disease (ILDs). Given the expertise of hospitals participating in EMPIRE, external central radiological or histopathological review were not considered necessary. IPF patients were always enrolled into EMPIRE at the discretion of the treating physician, but only when the diagnostic work-up including HRCT, lung biopsy if indicated, and MDT diagnosis was completed.

**Table 1. Baseline characteristics of patients with pirfenidone and patients on no antifibrotic treatment.** Data are given as mean (±SD) or N (%).

| | | Total N = 1626 | Pirfenidone N = 808 | No antifibrotic treatment N = 818 | P |
|---|---|---|---|---|---|
| Demographics | Men | 1 153 (70.9%) | 609 (75.4%) | 544 (66.5%) | < 0.001 |
| | Age (years) | 68.6 (±9.6) | 67.7 (±8.8) | 69.5 (±10.2) | < 0.001 |
| | BMI | 28.1 (±4.4) | 28.4 (±4.4) | 27.7 (±4.3) | 0.001 |
| Smoking | Never-smokers | 598 (36.9%) | 289 (35.8%) | 309 (38.0%) | 0.066 |
| | Ex-smokers | 227 (14.0%) | 101 (12.5%) | 126 (15.5%) | |
| | Current smokers | 796 (49.1%) | 418 (51.7%) | 378 (46.5%) | |
| HRCT pattern | Definite UIP | 1 096 (67.4%) | 582 (72.0%) | 514 (62.8%) | < 0.001 |
| | Possible UIP | 453 (27.9%) | 191 (23.6%) | 262 (32.0%) | |
| | Inconsistent with UIP | 70 (4.3%) | 35 (4.3%) | 35 (4.3%) | |
| | Not performed | 7 (0.4%) | 0 (0.0%) | 7 (0.9%) | |
| Histopathology | UIP | 161 (9.9%) | 111 (13.7%) | 50 (6.1%) | < 0.001 |
| | Probable UIP | 64 (3.9%) | 41 (5.1%) | 23 (2.8%) | |
| | Possible UIP | 58 (3.6%) | 32 (4.0%) | 26 (3.2%) | |
| | Not UIP | 38 (2.3%) | 14 (1.7%) | 24 (2.9%) | |
| | Not performed | 1 305 (80.3%) | 610 (75.5%) | 695 (85.0%) | |
| IPF diagnosis | IPF | 1 162 (71.5%) | 627 (77.6%) | 535 (65.4%) | < 0.001 |
| | Probable + possible IPF | 368 (22.6%) | 140 (17.3%) | 228 (27.9%) | |
| | Not IPF | 89 (5.5%) | 41 (5.1%) | 48 (5.9%) | |
| | Not performed | 7 (0.4%) | 0 (0.0%) | 7 (0.9%) | |
| Comorbidities | Number of comorbidities | 3.41 (1.99) | 3.58 (2.01) | 3.24 (1.95) | < 0.001 |
| | Heart and vascular | 1 150 (70.7%) | 596 (73.8%) | 554 (67.7%) | 0.007 |
| | Pulmonary | 483 (29.7%) | 251 (31.1%) | 232 (28.4%) | 0.233 |
| | Gastrointestinal | 863 (53.1%) | 483 (59.8%) | 380 (46.5%) | < 0.001 |
| | Urogenital | 259 (15.9%) | 133 (16.5%) | 126 (15.4%) | 0.560 |
| | Cancer | 94 (5.8%) | 45 (5.6%) | 49 (6.0%) | 0.716 |
| IPF treatment | Pharmacological | 1 018 (64.1%) | 808 (100.0%) | 210 (26.9%) | < 0.001 |
| | Rehabilitation | 256 (16.2%) | 191 (23.7%) | 65 (8.4%) | < 0.001 |
| | LTOT | 299 (18.9%) | 170 (21.1%) | 129 (16.6%) | 0.022 |
| | Lung transplantation | 146 (9.2%) | 103 (12.8%) | 43 (5.5%) | < 0.001 |
| Lung functions at baseline ± 3 months | FVC predicted (%) | 79.4 (±21.5) / 1,274[1] | 73.9 (±16.1) / 553[1] | 83.6 (±24.0) / 721[1] | < 0.001 |
| | DLCO predicted (%) | 49.2 (±19.1) / 1,188[1] | 46.7 (±14.5) / 518[1] | 51.2 (±21.8) / 670[1] | 0.008 |
| GAP index | I | 581 (45.9%) | 258 (41.6%) | 323 (50.1%) | 0.008 |
| | II | 553 (43.7%) | 289 (46.6%) | 264 (40.9%) | |
| | III | 131 (10.4%) | 73 (11.8%) | 58 (9.0%) | |
| Dyspnoea | NYHA I | 100 (8.7%) | 29 (4.7%) | 71 (13.2%) | < 0.001 |
| | NYHA II | 597 (51.8%) | 335 (54.6%) | 262 (48.6%) | |
| | NYHA III | 432 (37.5%) | 236 (38.4%) | 196 (36.4%) | |
| | NYHA IV | 24 (2.1%) | 14 (2.3%) | 10 (1.9%) | |

[1] Number of patients for whom the baseline value of FVC predicted or DLCO predicted was available

At baseline, participating IPF patients were assessed with HRCT to define UIP, possible UIP, and inconsistent with UIP; and histopathology, if indicated, to define UIP, probable UIP, possible UIP, and absence of (= no) UIP. Based on the combination of HRCT and lung biopsy findings, the final IPF diagnosis (definitive IPF, probable/ possible IPF, not IPF) was determined. The combination of HRCT and lung biopsy findings defined the subgroups for analysis (see Table 1).

## Outcomes

We report outcomes (OS, PFS and FVC decline) for the pirfenidone group and the no antifibrotic treatment group, and compare patients according to subgroups of diagnostic certainty. Short-term (lung function decline) and long-term (OS, PFS) outcomes of treatment were evaluated. Short-term outcomes included the change of FVC and DLCO for the first 12 months of follow-up. Long-term outcomes included the progression of disease or death from any cause. Progression of disease was defined present, when either a decline of FVC > 10%, or a decline of DLCO > 15%, or a decline of 6MWD > 50 m was observed, whichever came first, compared to baseline values. Progression free survival (PFS) and overall survival (OS) was calculated accordingly. Other parameters analysed at baseline only were GAP index [23] and dyspnoea rated according to the NYHA criteria [24].

## Statistical analysis

For continuous variables we report mean and standard deviation and for categorical variables absolute and relative frequency.

Kaplan-Meier methodology was used for the analysis of OS and PFS. Patients with no observed event during follow-up (death, lung transplantation or progression of disease, see definition above) were censored at the date of the last visit to the registry, the date of last documented pirfenidone treatment or the date of loss to follow-up, whichever came last.

To adjust for potential confounding, we used multivariate Cox proportional hazard models including covariates likely associated with outcomes OS and PFS. These covariates (age, sex, height, FVC at baseline and dyspnoea) were included into all Cox proportional hazard models. Using these Cox models, we then tested for interaction between pirfenidone (treatment *vs* no treatment) and the categories of diagnostic certainty (definitive, probable, possible).

Lung function in the first 12 months decline was analysed using a linear, mixed effects model. In this analysis we included only patients with a minimum of six months of follow up. The model on lung function decline was adjusted for age, height, sex, FVC at baseline and dyspnoea. The annual lung function decline was described by estimates of the time-dependent slope (including 95% CI). A random intercept and random time slope were included in the model specification. The model also tested for the significance of interaction between pirfenidone treatment and categories of diagnostic certainty. All statistical analysis was carried out using SPSS 25.0.0.0 and STATA 14.2. The level of significance α was set at 0.05.

## Results

### Patient characteristics at baseline

At baseline 1,626 IPF patients were included; 808 patients were treated with pirfenidone and 818 patients received no antifibrotic treatment. There was a higher frequency of visits in the patients treated with pirfenidone than in the patients without antifibrotic treatment (mean 3.3 vs 1.9 visits for the first 12 months of follow-up, and 5.3 vs 2.7 visits for the first 36 months of follow-up, respectively). The pirfenidone group included more men (75.4% vs 66.5%), was slightly younger (mean age 67.7 vs 69.5 years), included a smaller percentage of non-smokers (35.8% vs 38.0%), had higher body mass index (28.4 vs 27.7), and a less favourable GAP index (GAP 1: 41.6% vs. 50.1%) and dyspnoea distribution (NYHA I: 29 (4.7%) vs. 71 (13.2%); NYHA I + II: 59.3% vs 61.8%) at baseline. Further, patients in the pirfenidone group had lower FVC (absolute FVC mean 2.47 l vs 2.71 l; predicted FVC mean 73.9% vs 83.6%) and DLCO (predicted mean 46.7 vs 51.2%) at baseline.

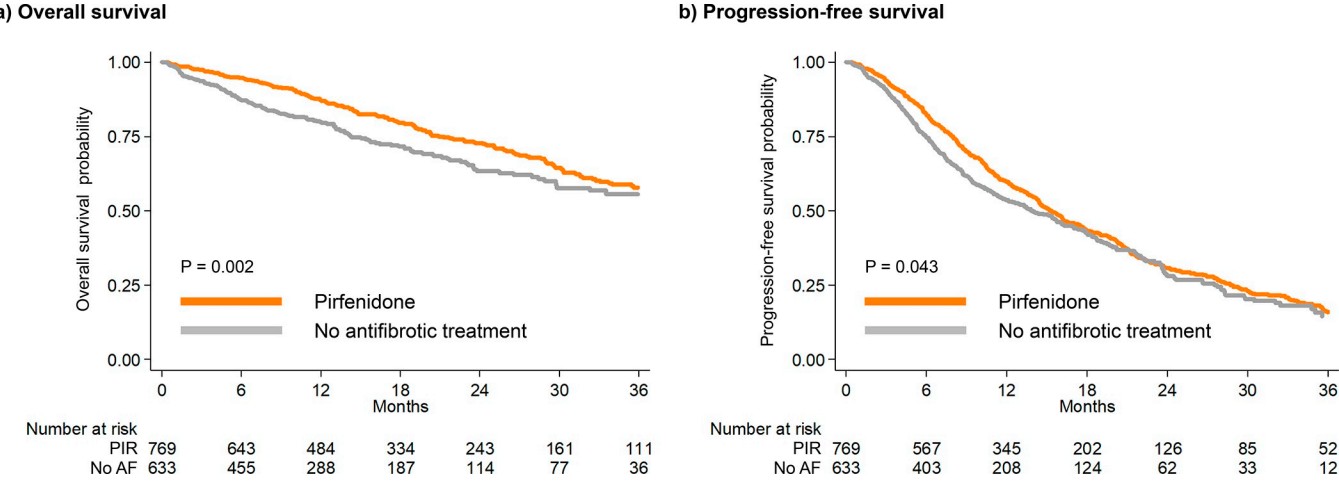

**Fig 1.** Overall survival (a) and progression-free survival (b) in the whole IPF cohort.

The most frequently reported comorbidities were cardiovascular (73.8% vs 67.7% of patients), gastrointestinal (59.8% vs 46.5%) and pulmonary (31.1% vs 28.4%) for the comparison between the pirfenidone and the no antifibrotic treatment group.

In patients treated with pirfenidone and stratified according to the diagnostic certainty of IPF, the proportion of men was highest in the group presenting with an UIP pattern on HRCT. Subgroups of different diagnostic certainty did not substantially differ with regard to FVC, DLCO, GAP index and dyspnoea.

Baseline characteristics of patients are presented in Table 1 (all patients) and in S1 Table (patients treated with pirfenidone stratified according to the certainty of IPF diagnosis). Due to the small number of patients with possible IPF according to the 2011 diagnostic criteria (7 in the pirfenidone group and 3 in the group with no antifibrotic treatment), the patients with possible IPF were combined with patients with probable IPF in data analysis.

## Overall survival and progression-free survival

In patients with IPF and regardless of the degree of diagnostic certainty, OS was greater in patients treated with pirfenidone compared to those not receiving antifibrotic treatment. For the two groups (pirfenidone vs no antifibrotic treatment) survival was 87.1% vs 79.8% after one year, 72.8% vs 63.2% after two years, and 57.9% vs 55.6% after three years of follow-up respectively (Fig 1A).

A similar pattern was observed for PFS. PFS for the two groups (pirfenidone vs no antifibrotic treatment) was 59.7% vs 53.6% after one year, 30.9% vs 28.1% after two years, and 15.8% vs 14.5% after three years of follow-up, respectively (Fig 1B). Median time to progression of IPF was 15.3 months for the pirfenidone and 13.9 months for the no antifibrotic treatment group.

In multivariate Cox proportional hazard models, co-variates sex, FVC and dyspnoea at baseline were found significantly associated with both OS and PFS (S2 and S3 Tables). All covariates were kept in following models to adjust for potential confounding. For adjusted analysis, treatment with pirfenidone was significantly associated with better OS (HR 0.749, 95% CI 0.575 to 0.976; P = 0.032), indicating a 25.1% reduction in mortality. However, no association was observed when PFS was analysed following adjustment (1.037, 95% CI 0.880 to 1.222; P = 0.666), see Table 2.

**Table 2. Hazard ratios associated with mortality and progression of IPF comparing pirfenidone with no antifibrotic treatment according to diagnostic subgroups (adjusted for age, sex, height, NYHA and FVC at baseline).**

| | | No. of patients | Mortality HR (95% CI) | P[1] | P for interaction[2] | Progression HR (95% CI) | P[1] | P for interaction[2] |
|---|---|---|---|---|---|---|---|---|
| Total | | 1,128 | 0.749 (0.575; 0.976) | **0.032** | | 1.037 (0.880; 1.222) | 0.666 | |
| HRCT | UIP | 735 | 0.668 (0.492; 0.906) | **0.010** | 0.579 | 0.845 (0.696; 1.027) | 0.090 | 0.109 |
| | Possible UIP | 342 | 0.765 (0.465; 1.259) | 0.292 | | 1.198 (0.892; 1.610) | 0.229 | |
| | Inconsistent with UIP | 51 | 0.418 (0.150; 1.161) | 0.094 | | 0.749 (0.386; 1.452) | 0.392 | |
| Histopathology | UIP | 119 | 0.495 (0.218; 1.122) | 0.092 | 0.578 | 0.647 (0.369; 1.134) | 0.129 | 0.497 |
| | Probable UIP | 52 | 0.449 (0.140; 1.436) | 0.177 | | 0.624 (0.306; 1.272) | 0.194 | |
| | Possible UIP | 40 | 0.345 (0.088; 1.347) | 0.126 | | 0.618 (0.275; 1.387) | 0.243 | |
| | Not UIP | 28 | - | - | | 1.560 (0.530; 4.589) | 0.419 | |
| IPF diagnosis | IPF | 787 | 0.653 (0.486; 0.876) | **0.005** | 0.401 | 0.850 (0.703; 1.028) | 0.094 | 0.194 |
| | Probable + possible IPF | 267 | 0.798 (0.462; 1.378) | 0.418 | | 1.189 (0.859; 1.646) | 0.297 | |
| | No | 74 | 0.335 (0.106; 1.053) | 0.061 | | 0.985 (0.531; 1.829) | 0.963 | |

[1] difference pirfenidone vs no antifibrotic treatment

[2] differential effect of pirfenidone between diagnostic subgroups; higher P value (above 0.05) indicates that the possible effect of pirfenidone therapy (a difference between pirfenidone and no antifibrotic treatment) is similar across all diagnostic subgroups

## Overall survival and progression-free survival according to diagnostic certainty of IPF

Overall survival was higher in patients treated with pirfenidone in all subgroups according to the diagnostic certainty (statistically significant results were observed namely for HRCT–UIP and final IPF diagnosis). Our analysis with multivariate model did not demonstrate a different association between pirfenidone treatment and OS for the different diagnostic IPF subgroups, i.e. definitive, and probable/possible IPF (Table 2, P value for interaction > 0.40 for each of the three classifications); although there seems to be a less pronounced difference in crude survival curves for some subgroups (S2 Fig). OS in patients treated with pirfenidone was similar across all diagnostic subgroups (S3 Fig).

Although the unadjusted comparison suggested a possible effect of pirfenidone on PFS (S4 Fig), also observed among subgroups (S5 Fig), this was not confirmed by adjusted multivariate analysis (Table 2).

## Lung function decline in first 12 months

The FVC decline was -0.073 l/year (95% CI, -0.124 to -0.023 l/year) in the pirfenidone group and -0.169 l/year (95% CI, -0.230 to -0.109 l/year) in the group not receiving antifibrotic treatment (P = 0.017) (Table 3). On the other hand, no significant difference was observed for DLCO decline: -0.187 mmol/kPa/min (95% CI, -0.396 to -0.023 mmol/kPa/min) in the pirfenidone group, -0.116 mmol/kPa/min (95% CI, -0.355 to -0.122 mmol/kPa/min) in the no antifibrotic treatment group (P = 0.665) (S4 Table).

## Lung function decline in first 12 months according to diagnostic certainty of IPF

The effect of pirfenidone to slow FVC decline was most significant in patients with a UIP pattern on HRCT (-0.078 l/yr for the pirfenidone group, 95% CI, -0.140 to -0.016 l/yr; -0.209 l/yr for the no antifibrotic treatment group, 95% CI, -0.289 to -0.130 l/yr; P = 0.011) and in patients

**Table 3. Difference of annual FVC decline according to diagnostic subgroups (adjusted for age, sex, height, NYHA and absolute FVC at baseline).**

| | | No. of patients (pirfenidone) | FVC decline annual rate–pirfenidone (95% CI) (l/yr) | No. of patients (no antifibrotics) | FVC decline annual rate–no antifibrotic treatment (95% CI) (l/yr) | P[1] | P value for interaction[2] |
|---|---|---|---|---|---|---|---|
| Total | | 526 | -0.073 (-0.124; -0.023) | 454 | -0.169 (-0.230; -0.109) | **0.017** | |
| HRCT | UIP | 361 | -0.078 (-0.140; -0.016) | 275 | -0.209 (-0.289; -0.130) | **0.011** | **0.047** |
| | Possible UIP | 142 | -0.040 (-0.137; 0.057) | 158 | -0.138 (-0.236; -0.041) | 0.160 | |
| | Inconsistent with UIP | 23 | -0.181 (-0.410; 0.048) | 21 | 0.061 (-0.212; 0.334) | 0.183 | |
| Histopathology | UIP | 71 | -0.095 (-0.210; 0.021) | 28 | -0.163 (-0.374; 0.047) | 0.575 | 0.178 |
| | Probable UIP | 28 | -0.110 (-0.293; 0.073) | 15 | 0.138 (-0.111; 0.388) | 0.115 | |
| | Possible UIP | 23 | 0.017 (-0.178; 0.212) | 14 | -0.144 (-0.369; 0.081) | 0.288 | |
| | Not UIP | 9 | 0.086 (-0.238; 0.411) | 17 | -0.116 (-0.336; 0.104) | 0.312 | |
| IPF diagnosis | IPF | 393 | -0.073 (-0.132; -0.015) | 285 | -0.193 (-0.272; -0.114) | **0.017** | 0.513 |
| | Probable + possible IPF | 101 | -0.078 (-0.191; 0.034) | 144 | -0.167 (-0.272; -0.062) | 0.261 | |
| | Not IPF | 32 | -0.072 (-0.300; 0.155) | 25 | -0.003 (-0.213; 0.208) | 0.660 | |

[1] difference pirfenidone vs no antifibrotic treatment

[2] differential effect of pirfenidone between diagnostic subgroups; higher P value (above 0.05) indicates that the possible effect of pirfenidone therapy (a difference between pirfenidone and no antifibrotic treatment) is similar across all diagnostic subgroups

with confirmed IPF (-0.073 l/yr for the pirfenidone group, 95% CI -0.132 to -0.015 l/yr; -0.193 l/yr for the no antifibrotic treatment group; 95% CI -0.272 to -0.114; P = 0.017) (Table 3). We found a borderline statistically significant difference in FVC decline according to HRCT subgroups and in relation to pirfenidone treatment (P = 0.047), suggesting a potentially stronger effect of pirfenidone in patients with definite UIP/IPF.

No significant difference was observed for DLCO decline within diagnostic subgroups, irrespective of treatment with pirfenidone (S4 Table).

## Discussion

Using follow-up data from the multi-national EMPIRE registry, we compared diagnostic subgroups of IPF patients as defined by the 2011 ATS/ERS/JRS/ALAT criteria. The clinical characteristics and course of IPF for patients receiving pirfenidone were compared to those of patients on no antifibrotic treatment. Our results are in line with randomized clinical trials and other real-world data, strongly indicating that patients with IPF profit from pirfenidone, experiencing longer OS and smaller FVC decline [1, 2, 6, 7].

Our results further suggest that the effect of pirfenidone on these outcomes was not significantly different between subgroups of IPF as defined by diagnostic certainty. Patients treated with pirfenidone in EMPIRE were found similar to other real-life cohorts as reported in literature (in terms of age and sex) However, FVC % predicted at baseline was rather lower compared to other real-life studies [13, 15, 17, 25–27].

In the literature, OS for patients treated with pirfenidone varies with 1-year OS between 85% and 99% [15, 16, 25], 2-year OS between 76.9% and 83% [16, 17], and 3-year OS between 73% and 74% [15, 26]. The OS in EMPIRE were similar or slightly lower (87.1%, 72.8% and 57.9% for one-, two, and three-year follow-up). Median time to death was therefore not reached during 36-month follow-up.

Differences in progression-free survival for the pirfenidone group and the no antifibrotic treatment group were not as apparent as for the overall survival. The Kaplan-Meier analysis

indicates an effect lasting approximately 12 months. A more pronounced impact of antifibrotic treatment on OS rather than on lung function decline was recently reported from the German IPF registry study [18]. A higher frequency of visits in the pirfenidone group may also have increased the probability of detecting progression of IPF earlier.

Nevertheless, some IPF patients were lost to follow-up, the proportion being markedly higher for the group receiving no antifibrotic treatment, likely resulting into more favourable effect estimates for the group with no antifibrotic treatment, than would have been observed provided all the patients stayed in the study [12]. This differential loss to follow-up could have caused underestimation of the difference for PFS and lung function later on, when mainly patients with better health status at baseline remained alive. This was also illustrated in the comparison of baseline characteristics of patients completing 24 months of follow-up [28]. The median time to progression of patients treated with pirfenidone in our study is comparable with data published in literature [16, 27].

The effect of pirfenidone on OS was most clearly seen in the group of patients with the highest diagnostic certainty, i.e. those with a radiological and/or histopathological pattern of definite UIP and/or a final IPF diagnosis. This finding can be attributed to the much higher number of patients in this groups compared to IPF patients with a lower diagnostic certainty. Nevertheless, our analysis indicates that the effect of pirfenidone is not significantly different between the diagnostic IPF subgroups, suggesting that patients with probable and possible UIP/IPF might also profit from the pirfenidone treatment.

To our knowledge, this is the first registry-based study including real-world patients specifically focusing on differences in treatment outcomes related to the degree of diagnostic certainty of IPF. Long-term follow-up of the large real-life EMPIRE cohort allowed us to illustrate these relationships; however, quality of data needs to be taken into account when interpreting the data. The EMPIRE data have been collected in nearly 50 centres from 11 countries; this sampling could have caused increased between-site variability, and therefore lead to underestimation of study results. Furthermore, given that this is a patient registry and not a randomised clinical trial, a number of sources of bias and uncertainties should be considered (healthy participation bias, different baseline patient characteristics, rules for drug administration, different time order of diagnosis, admission visit and treatment initiation, etc.).

Some limitations of this analysis based on data from the EMPIRE registry might arise from its real-life nature and multicentre data collection. In the majority of cases, diagnosis of IPF is based on the assessment of HRCT images, which can be subjective. IPF diagnosis was determined in each participating centre without central reading or re-assessment. Although the patients with definite UIP/IPF prevail in the analysed cohort, there is also a sufficiently large cohort of patients with HRCT findings of lower certainty (probable and possible UIP) available for the comparison.

Management of IPF in clinical practice faces several limitations arising from disease characteristics and real-world settings: IPF is a rare disease and there is significant loss to follow-up of patients with less favourable prognosis and baseline characteristics, particularly when they do not receive antifibrotics. These patients are less prone to travel and usually receive palliative care at home provided by their general practitioners. On the other hand, patients with better baseline characteristics remain under follow-up for longer time periods, which may lead to the conclusion that health status has improved over time (e.g. in terms of greater FVC or less dyspnoea reported). This type of bias was illustrated in our previous study, in which baseline characteristics of patients staying at risk (under follow-up) for 6, 12, 18, and 24 months were compared–numbers of patients at risk decreased over time and the survivors were characterised by a higher FVC and DLCO, and less advanced dyspnoea at baseline, particularly in the group on no antifibrotic treatment [28]. The effect of drop-out results in generally low

numbers of patients available for analysis not so much at diagnosis or treatment initiation, but mainly after a longer follow-up period (e.g. 3 years and more), when the remaining group of patients may have different characteristics than the original cohort due to substantial and selective drop out [12]. The real-world IPF studies published to date are usually based on dozens or lower hundreds of patients and follow-up is two or three years [13, 15–18, 25, 27], although some long-term data from large national or international IPF registries are available as well [29, 30].

It may be challenging to define a reasonable baseline visit in a retrospective registry-based study. Patients may be included in the registry at different times after diagnosis (or even before), and before or after treatment is started (both antifibrotic and non-antifibrotic). In this study, the baseline was set at pirfenidone therapy initiation for the pirfenidone group and admission visit (enrolment) for the no antifibrotic treatment group. The pirfenidone group and the no antifibrotic treatment group did not differ significantly in terms of the time pattern from diagnosis over admission visit to treatment initiation, although the analysed cohort included patients diagnosed before the defined period for enrolment (2015–2018). Mean and median time from diagnosis to admission visit was 9.81 and 0.41 months for the pirfenidone group and 13.57 and 0.82 months for the no antifibrotic treatment group, respectively. Mean and median time from the admission visit to the pirfenidone therapy initiation was 2.52 and 1.16 months, respectively.

In conclusion, this analysis of real-life data from the international EMPIRE registry confirmed favourable clinical outcomes associated with pirfenidone treatment. Treatment with pirfenidone was associated increased OS and a slower deterioration of lung function. This benefit of pirfenidone is likely present in IPF patients irrespective of their degree of diagnostic certainty.

## Supporting information

**S1 Fig. Flow diagram of patients participating in the study and their availability for performed analyses.**
(PDF)

**S2 Fig. Overall survival in diagnostic subgroups.**
(PDF)

**S3 Fig. Overall survival of patients treated with pirfenidone in diagnostic subgroups.**
(PDF)

**S4 Fig. Progression-free survival in diagnostic subgroups.**
(PDF)

**S5 Fig. Progression-free survival of patients treated with pirfenidone in diagnostic subgroups.**
(PDF)

**S1 Appendix. Dataset for Fig 1.**
(XLSX)

**S2 Appendix. Abstract from the ATS conference.** Pirfenidone Effectiveness In Idiopathic Pulmonary Fibrosis With Different Radiologic Patterns.
(PDF)

**S1 Table. Baseline characteristics of patients treated with pirfenidone according to diagnostic subgroups.**
(PDF)

**S2 Table. Univariate hazard ratio for mortality and PFS.**
(PDF)

**S3 Table. Multivariate hazard ratio for mortality and PFS (adjusted on age, sex, height, NYHA and FVC at baseline).**
(PDF)

**S4 Table. Difference of annual rate of decline in DLCO according to diagnostic subgroups.**
(PDF)

## Acknowledgments

The authors thank investigators in clinical centres involved in the EMPIRE project for their valuable collaboration and data acquisition.

## Author Contributions

**Conceptualization:** Ladislav Dušek, Martina Koziar Vašáková.

**Data curation:** Nesrin Mogulkoć, Katarzyna Lewandowska, Martina Šterclová, Veronika Müller, Marta Hájková, Mordechai R. Kramer, Jasna Tekavec-Trkanjec, Dragana Jovanović, Michael Studnicka, Natalia Stoeva.

**Formal analysis:** Ondřej Májek, Simona Littnerová, Ladislav Dušek.

**Funding acquisition:** Klaus-Uwe Kirchgässler, Ladislav Dušek.

**Methodology:** Ondřej Májek, Simona Littnerová, Martina Koziar Vašáková.

**Project administration:** Jakub Gregor.

**Resources:** Ladislav Dušek.

**Supervision:** Michael Studnicka, Martina Koziar Vašáková.

**Writing – original draft:** Ondřej Májek, Jakub Gregor, Martina Koziar Vašáková.

**Writing – review & editing:** Nesrin Mogulkoć, Katarzyna Lewandowska, Martina Šterclová, Veronika Müller, Marta Hájková, Mordechai R. Kramer, Jasna Tekavec-Trkanjec, Dragana Jovanović, Michael Studnicka, Natalia Stoeva, Klaus-Uwe Kirchgässler.

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
