## [Decision Letter · Decision Letter 0]

16 Feb 2022

PONE-D-21-27861Survival and lung function decline in patients with definite, probable and possible idiopathic pulmonary fibrosis treated with pirfenidonePLOS ONE

Dear Dr. Vasakova,

Thank you for submitting your manuscript to PLOS ONE. After careful consideration, we feel that it has merit but does not fully meet PLOS ONE’s publication criteria as it currently stands. Therefore, we invite you to submit a revised version of the manuscript that addresses the points raised during the review process.Please ensure that your decision is justified on PLOS ONE’s publication criteria and not, for example, on novelty or perceived impact.

We look forward to receiving your revised manuscript.

Kind regards,

Minghua Wu, M.D., Ph.D.

Academic Editor

PLOS ONE

https://journals.plos.org/plosone/s/fileid=ba62/PLOSOne_formatting_sample_title_authors_affiliations.pdf".

“This study was supported by F. Hofmann-La Roche. The EMPIRE registry is supported by Boehringer Ingelheim and F. Hofmann-La Roche.”

“This study was supported by F. Hofmann-La Roche. The EMPIRE registry is supported by Boehringer Ingelheim and F. Hofmann-La Roche.

The authors thank investigators in clinical centres involved in the EMPIRE project for their valuable collaboration and data acquisition.”

“This study was supported by F. Hofmann-La Roche. The EMPIRE registry is supported by Boehringer Ingelheim and F. Hofmann-La Roche.”

“Dr. Kirchgässler is an employee of F. Hoffmann-La Roche Ltd.

Dr. Lewandowska reports grants, personal fees and non-financial support from F. Hofmann-La Roche, grants, personal fees and non-financial support from Boehringer Ingelheim.

Dr. Stoeva reports and received fees from Boehringer Ingelheim for participation in advisory boards and from Roche for presentations.

Dr. Studnicka reports grants and personal fees from Boehringer-Ingelheim, grants and personal fees from Roche, during the conduct of the study.

Dr. Tekavec-Trkanjec reports personal fees from Roche, personal fees and other from Boehringer Ingelheim.”

5. We noted in your submission details that a portion of your manuscript may have been presented or published elsewhere. [Some of the results were presented in a form of electronic poster at the ATS Congress 2020.] Please clarify whether this [conference proceeding or publication] was peer-reviewed and formally published. If this work was previously peer-reviewed and published, in the cover letter please provide the reason that this work does not constitute dual publication and should be included in the current manuscript.

Reviewers' comments:

Reviewer's Responses to Questions

**Comments to the Author**

1. Is the manuscript technically sound, and do the data support the conclusions?

Reviewer #1: Partly

2. Has the statistical analysis been performed appropriately and rigorously? 

Reviewer #1: Yes

3. Have the authors made all data underlying the findings in their manuscript fully available?

Reviewer #1: Yes

4. Is the manuscript presented in an intelligible fashion and written in standard English?

Reviewer #1: Yes

5. Review Comments to the Author

Reviewer #1: This study focuses on the outcomes of the pirfenidone-treated group and the untreated group registered in the EMPIRE Registry. The study also compared the definite IPF with probable/possible UIP, and examined the differences in prognosis and changes in lung function depending on the degree of certainty of the UIP. The pirfenidone-treated group had better OS and PFS than the untreated group, but the multivariate results were significantly better only for OS, which was consistent with the previous report.

The authors further showed that the effect of pirfenidone is the same as that of UIP even with probable UIP pattern.

These results are consistent with previous studies, and since probable UIP is considered to be UIP in the current guidelines, examining the effect of pirfenidone treatment depending on the degree of UIP is a less important issue.

However, as real-world data, it is commendable that the pirfenidone-treated group had a better prognosis despite significantly lower baseline lung function than the untreated group.

Since we do not have to pay attention to the degree of UIP pattern in the antifibrotic treatment, it is a proposal to reconsider the title of this paper.

1) Since probable/possible UIP are diagnosed as UIP in the latest guideline, the title of this manuscript is of little appeal to physicians managing IPF treatment.

2) In this study, how are patients who have been discontinued from pirfenidone treatment or who have been switched from nintedanib to pirfenidone?

3) mention more deeply the difference from nintedanib treatment and how the results of this study affect when you are confused with nintedanib. There is a need to

4) Since nintedanib is indicated for PF-ILD, the chances of introducing pirfenidone are decreasing currently. The significance of the results of this study on the treatment of pirfenidone in such situations should be discussed more.

5) Do “other treatments” in Tables and Figures refer to “no antifibrotic treatment”? Does it also include nintedanib treatment as other treatment? It's easy to get confused, so it's better to unify them so that they are easy to understand.

6) Please describe the difference in the onset of acute exacerbations between the pirfenidone group and the untreated group. It's very important to discuss why the survival curves are different between these groups in this study. It is plausible that the prolongation effect of OS cannot be explained only by suppressing the decline in FVC by pirfenidone treatment.

7) In the section of the results, “Overall survival and progression-free survival according to diagnostic certainly of IPF”, the authors should explain the all data of the Figures in detail.

You mentioned that there was no significant different effect of pirfenidone treatment on OS for the different diagnostic IPF subgroups in Table 2 and Sup. Fig. 2. In Table 2, it seems that there was a difference between diagnostic certainly of IPF distinct from Sup Fig. 2. It was hard to understand your results for general readers.

Further, you described the OS in patients treated with pirfenidone regarding Sup. Fig. 3. However, Sup. Fig. 3 demonstrates PFS, but not OS, in diagnostic subgroups.

Moreover, I cannot find Sup. Fig. 5 in the supplementary file.

Please specify the definition of p1 and p2 in the Tables. Your presentation throughout the results including table capture is unfriendly to the readers.

6. PLOS authors have the option to publish the peer review history of their article (what does this mean?). If published, this will include your full peer review and any attached files.

Reviewer #1: **Yes: **Yasuhiko Koga

---

## [Author Response · Author response to Decision Letter 0]

24 Mar 2022

Dear Dr. Minghua Wu,

We would like to thank you and the reviewer for the comprehensive review of our article. We have made changes to reflect readability and clarity of the paper as suggested.

Editors' comments

We modified and transferred information about funding and acknowledgements as requested. We explained connection to the previous publication (ATS poster) in the cover letter.

Please note that a new Conflict of Interest statement of Veronika Müller has been added.

Reviewers' comments

1) Since probable/possible UIP are diagnosed as UIP in the latest guideline, the title of this manuscript is of little appeal to physicians managing IPF treatment.

We agree that the stratification according to the diagnostic certainty with regard to applied treatment has currently become of lower importance. However, our paper demonstrates that this practice is justifiable and that the therapy likely works in all groups. Such evidence has been missing for pirfenidone. We therefore prefer to maintain the current title.

2) In this study, how are patients who have been discontinued from pirfenidone treatment or who have been switched from nintedanib to pirfenidone?

Described in Methods: “Patients followed in EMPIRE and receiving nintedanib at any time were excluded from analysis (see S1 Fig 1).” The “control” group contains only patients with no history of antifibrotic treatment. Figure legends and tables were corrected from “other treatment” to “no antifibrotic treatment” to be consistent with the rest of the paper.

3) mention more deeply the difference from nintedanib treatment and how the results of this study affect when you are confused with nintedanib. There is a need to

If the reviewer refers to switch from/to nintedanib, we agree that the nintedanib phase should not be included. This is why we excluded patients with previous nintedanib therapy from the analysis (since the final therapy outcome might be affected).

4) Since nintedanib is indicated for PF-ILD, the chances of introducing pirfenidone are decreasing currently. The significance of the results of this study on the treatment of pirfenidone in such situations should be discussed more.

This is a correct point, but we feel that it falls out of scope of our paper a bit. Furthermore, prescription of both antifibrotics differs from country to country, even within the EMPIRE registry; we therefore assume that pirfenidone remains an important option, at least for patients who do not tolerate the nintedanib therapy well.

5) Do “other treatments” in Tables and Figures refer to “no antifibrotic treatment”? Does it also include nintedanib treatment as other treatment? It’s easy to get confused, so it’s better to unify them so that they are easy to understand.

Yes, thank you, legends have been corrected as explained above.

6) Please describe the difference in the onset of acute exacerbations between the pirfenidone group and the untreated group. It’s very important to discuss why the survival curves are different between these groups in this study. It is plausible that the prolongation effect of OS cannot be explained only by suppressing the decline in FVC by pirfenidone treatment.

This is a reasonable assumption, but we found no difference in the onset of acute exacerbations between the pirfenidone and no antifibrotics group when analysing this dataset (median time to the first AE 5.38 months for PIR vs 5.48 months for no AF, P = 0.509). However, overall frequency of exacerbations was relatively low (60 events in the entire cohort of 1,600 patients).

7) In the section of the results, “Overall survival and progression-free survival according to diagnostic certainly of IPF”, the authors should explain all data of the Figures in detail.

We have added a brief interpretation and summary under each Figure in the supplementary material.

You mentioned that there was no significant different effect of pirfenidone treatment on OS for the different diagnostic IPF subgroups in Table 2 and Sup. Fig. 2. In Table 2, it seems that there was a difference between diagnostic certainly of IPF distinct from Sup Fig. 2. It was hard to understand your results for general readers.

Hopefully the interpretation under the supplementary figures helps. We also modified the main text referring to these outputs – Table 2 is an adjusted analysis in which differences between pirfenidone and no antifibrotics are assessed, while Suppl. Fig. 3 (originally Suppl. Fig. 2 in the reviewed version of the paper, see below) is an unadjusted analysis comparing pirfenidone patients only, without any involvement of patients with no antifibrotics.

Further, you described the OS in patients treated with pirfenidone regarding Sup. Fig. 3. However, Sup. Fig. 3 demonstrates PFS, but not OS, in diagnostic subgroups.

Moreover, I cannot find Sup. Fig. 5 in the supplementary file.

Apology for this inconvenience – the author responsible for the preparation of figures and supplementary material forgot to include Figure 1 in the supporting info document (a flow diagram of patients included in the analysis). Numbering of all supplementary figures therefore moved by one. The revised version includes the correct supplementary Figure 1 and numbering has been corrected.

Please specify the definition of p1 and p2 in the Tables. Your presentation throughout the results including table capture is unfriendly to the readers.

Added in headings and footnotes for better clarity.

---

## [Decision Letter · Decision Letter 1]

10 Jun 2022

PONE-D-21-27861R1Survival and lung function decline in patients with definite, probable and possible idiopathic pulmonary fibrosis treated with pirfenidonePLOS ONE

Dear Dr. Vasakova,

Thank you for submitting your manuscript to PLOS ONE. After careful consideration, we feel that it has merit but does not fully meet PLOS ONE’s publication criteria as it currently stands. Therefore, we invite you to submit a revised version of the manuscript that addresses the points raised during the review process.

We look forward to receiving your revised manuscript.

Kind regards,

Minghua Wu, M.D., Ph.D.

Academic Editor

PLOS ONE

Journal Requirements:

Reviewers' comments:

Reviewer's Responses to Questions

**Comments to the Author**

1. If the authors have adequately addressed your comments raised in a previous round of review and you feel that this manuscript is now acceptable for publication, you may indicate that here to bypass the “Comments to the Author” section, enter your conflict of interest statement in the “Confidential to Editor” section, and submit your "Accept" recommendation.

Reviewer #1: All comments have been addressed

Reviewer #2: (No Response)

2. Is the manuscript technically sound, and do the data support the conclusions?

Reviewer #1: Yes

Reviewer #2: Partly

3. Has the statistical analysis been performed appropriately and rigorously? 

Reviewer #1: Yes

Reviewer #2: Yes

4. Have the authors made all data underlying the findings in their manuscript fully available?

Reviewer #1: Yes

Reviewer #2: Yes

5. Is the manuscript presented in an intelligible fashion and written in standard English?

Reviewer #1: Yes

Reviewer #2: No

6. Review Comments to the Author

Reviewer #1: (No Response)

Reviewer #2: (No Response)

7. PLOS authors have the option to publish the peer review history of their article (what does this mean?). If published, this will include your full peer review and any attached files.

Reviewer #1: **Yes: **Yasuhiko Koga

Reviewer #2: No

---

## [Author Response · Author response to Decision Letter 1]

24 Jun 2022

We would like to thank the reviewer for the comprehensive review of our article. We included the suggested corrections and changes, and we think that the overall readability and clarity of the paper has been significantly improved.

Please see our answers and explanations below.

Sincerely

Martina Koziar Vašáková

Revisions

1. Methods: “Patients with no observed event during follow-up (death, lung transplantation or progression of disease, see definition above) were censored at the date of the last visit to the registry, the date of last documented pirfenidone treatment or the date of loss to follow-up, whichever came last.” - It would be of value to know how patients that were started on pirfenidone and had to discontinue the medication early due to side effects were handled in the analysis. Eg. if patient was on pirfenidone for 1 month and then had to discontinue, was this patient still included in the pirfenidone arm? If there was a large proportion of patients like this, it would affect the interpretation of the results. Recommended addressing if this was considered and how it was handled, consider including the median duration of therapy (treatment with pirfenidone).

Patients treated with pirfenidone were always censored at the date of the PIR therapy termination. Our statement suggested that they were included in K-M even after this event, which was confusing. Thank you for this point, the text has been corrected accordingly.

2. Table 1: report standard deviations with �

Added

3. Table 1: BMI and % of non-smokers reported in table do not match with what is written in the body of text.

Corrected in text, thank you

4. Table 1: in the FVC and DLCO category, does the number in parenthesis represents standard deviation? If so, consider including the � to make it clear. Currently, it is confusing because in the first column the % sign is in parenthesis (%).

Added

5. Table 1: under smoking, is the P value representing the comparison between non-smokers? If so, then it should be placed in line with this row and not spanning the 3 rows. Similarly for HRCT pattern, Histopathology, IPF diagnosis, GAP index and dyspnoea.

The P value refers to the overall proportion of all categories of the particular variable in a patient group. E. g. Never/ex/current smokers, or NYHA I, II, III, IV. The patient groups may therefore be similar e.g. in the proportion of NYHA III or IV (the difference would be statistically insignificant), but the stages I and II make the statistically significant difference between PIR and No AF.

6. Table 1: baseline FVC and DLCO do not match with the data on supplementary Table 4.

Since a standalone output was published at ERS 2021 focused on healthy survivor bias, we decided to add this reference (no. 28) and remove Supplementary Table 4. There is slightly different methodological approach there in comparison to this paper and it might introduce confusion as pointed out by the reviewer. This also answers to the item 11.

7. Supplementary fig 1 flow diagram has an error. The numbers under patients with pirfenidone therapy and IPF diagnosis do not add up to 808 and do not match with Table 1. Table 1 does contain appropriate numbers as they add up to n=808.

Corrected, thank you

8. Table 2 and Table 3: caption 2 is confusing and misleading. It reads: “higher P value (above 0.05) indicates that the difference between pirfenidone and no antifibrotic treatment is consistent across all diagnostic subgroups,” this implies that there is a difference between the subgroups when the P value is above 0.05). Consider: higher P value (above 0.05) indicates that the difference between pirfenidone and no antifibrotic treatment in the different subgroups is not statistically significant.

We actually do not refer to the difference between PIR and NoAF alone, but this P value indicates whether this difference is similar across the diagnostic groups. If this P value was lower than 0.05, it would mean that pirfenidone has stronger effect in some of the diagnostic groups than in another. This happened in FVC decline with a borderline statistical significance (Table 3 – HRCT). But we agree that this particular output is not easy to understand – we have tried to make the explanation under Tables 2 and 3 more clear.

9. Supp. Table 3. The total number of patients on pirfenidone and on no antifibrotic treatment are different compared to the ones on Table 3 (FVC decline), why is this? Did some patients not have DLCO recorded?

Yes, completeness of lung function data is different and one of the biggest issues that we have to handle when analyzing data from our registry 

10. Discussion, lines 271-272: For clarification, in which study was median time to death not reached?

In none of them, the same as in our study (all over 50% after 3-y f-up). Since the sentence might be confusing, we have changed it to “Median time to death was therefore not reached during 36-month follow-up.” (in our study)

11. Discussion, lines 312-314: is there a reference for this statement?

Addressed in item 6

12. Figure 2. The legend has been corrected to describe the groups as “pirfenidone group” and the “no antifibrotic treatment group”; however, under each individual image within figure 2, legend still reads PRI and OT. Same applies for Figure 1a,1b as well as S1 Fig 4. If keeping the abbreviation OT, please specify that it represents the no antifibrotic group.

Corrected to “No AF”, thank you

Grammatical/syntax corrections

Thank you, we have made the indicated corrections.

1. Methods: We followed patients in the multi-national European MultiPartner IPF Registry (EMPIRE) first seen in between 2015 and 2018.” – grammatical error, just remove “in” so that it reads first seen between 2015 and 2018.

2. Introduction: “One group of particular concern is IPF patients, who do not present with definite usual interstitial pneumonia (UIP) on HRCT, but demonstrate the radiological and/or histopathological pattern of probable/ possible UIP” - syntax error, remove comma after IPF patients

3. Results – Overall survival and progression-free survival according to diagnostic certainty of IPF: “although there seems to be less pronounced difference in crude survival curves for some subgroups (S1 Fig 2).” – this statement is missing an “a”- “although there seems to be a less pronounced difference in crude survival curves for some subgroups in the unadjusted survival analysis (S1 Fig 2).”

4. Results – Overall survival and progression-free survival according to diagnostic certainty of IPF: “Although the unadjusted comparison suggested a possible effect of pirfenidone on PFS (S1 Fig 4), with PFS in patients treated with pirfenidone rather similar among subgroups (S1 Fig 5), this was not confirmed by adjusted multivariate analysis (Table 2)” – This phrase can be made friendlier to the reader, consider -Although the unadjusted comparison suggested an a possible effect of pirfenidone on PFS (S1 Fig 4), also observed among subgroups (S1 Fig 5), this was not confirmed by adjusted multivariate analysis

5. Supplementary fig 1. Patients on/not on pirfenidone (or nintedanib) or patients not on pirfenidone (instead of with and without)

6. Line 87: should read: population, importance new evidence

7. Line 94: should read: January 1st 2015 and December 31st 2018 and followed through October 29th 2019.

8. Line 96: should read: patient included in the EMPIRE registry

9. Line 98: should read: indicated, and a MDT discussion for final…

10. Line 108: remove word “onwards”

11. Line 163: significance was set “at” instead of set “on”

12. Line 196: authors use a hyphen (-) in no-antifibrotic treatment. No hyphen is used throughout the rest of the article. Consider removing it to stay consistent throughout the text.

13. Line 258-260: “Patient’s characteristics and the course of IPF were analyzed for those, who were treated with pirfenidone and compared to those receiving no antifibrotic treatment.” – this sentence does not read well. Consider: The clinical characteristics and course of IPF for patients receiving pirfenidone were compared to those of patients on no antifibrotic treatment. 

14. Line 318: decrease in time � decrease over time

15. Line 320: patient with no antifibrotic treatment � patients on no antifibrotic treatment

16. Fig 1 demographics instead of demography

---

## [Decision Letter · Decision Letter 2]

17 Aug 2022

Survival and lung function decline in patients with definite, probable and possible idiopathic pulmonary fibrosis treated with pirfenidone

PONE-D-21-27861R2

Dear Dr. Vasakova,

We’re pleased to inform you that your manuscript has been judged scientifically suitable for publication and will be formally accepted for publication once it meets all outstanding technical requirements.

Kind regards,

James West, PhD

Academic Editor

PLOS ONE

Additional Editor Comments (optional):

Reviewers' comments:

Reviewer's Responses to Questions

**Comments to the Author**

1. If the authors have adequately addressed your comments raised in a previous round of review and you feel that this manuscript is now acceptable for publication, you may indicate that here to bypass the “Comments to the Author” section, enter your conflict of interest statement in the “Confidential to Editor” section, and submit your "Accept" recommendation.

Reviewer #1: All comments have been addressed

Reviewer #2: All comments have been addressed

2. Is the manuscript technically sound, and do the data support the conclusions?

Reviewer #1: Yes

Reviewer #2: Yes

3. Has the statistical analysis been performed appropriately and rigorously? 

Reviewer #1: Yes

Reviewer #2: Yes

4. Have the authors made all data underlying the findings in their manuscript fully available?

Reviewer #1: Yes

Reviewer #2: Yes

5. Is the manuscript presented in an intelligible fashion and written in standard English?

Reviewer #1: Yes

Reviewer #2: Yes

6. Review Comments to the Author

Reviewer #1: To the Authors,

The authors revised all of the comments of their manuscript.

All comments have been addressed

It is suitable for the publication of the PlosOne.

Reviewer #2: (No Response)

7. PLOS authors have the option to publish the peer review history of their article (what does this mean?). If published, this will include your full peer review and any attached files.

Reviewer #1: **Yes: **YASUHIKO KOGA

Reviewer #2: No

---

## [Editor Report · Acceptance letter]

22 Aug 2022

PONE-D-21-27861R2 

Survival and lung function decline in patients with definite, probable and possible idiopathic pulmonary fibrosis treated with pirfenidone 

Dear Dr. Vasakova:

I'm pleased to inform you that your manuscript has been deemed suitable for publication in PLOS ONE. Congratulations! Your manuscript is now with our production department. 

Kind regards, 

on behalf of

Dr. James West 

Academic Editor

PLOS ONE